# The Effect of Bovine Viral Diarrhea Virus (BVDV) Strains and the Corresponding Infected-Macrophages’ Supernatant on Macrophage Inflammatory Function and Lymphocyte Apoptosis

**DOI:** 10.3390/v12070701

**Published:** 2020-06-29

**Authors:** Karim Abdelsalam, Mrigendra Rajput, Gamal Elmowalid, Jacob Sobraske, Neelu Thakur, Hossam Abdallah, Ahmed A. H. Ali, Christopher C. L. Chase

**Affiliations:** 1Department of Veterinary and Biomedical Sciences, South Dakota State University, Brookings, SD 57007, USA; mrajput@atu.edu (M.R.); jacob.sobraske@sdstate.edu (J.S.); neelu.thakur@sdstate.edu (N.T.); 2Faculty of Veterinary Medicine, Zagazig University, Zagazig 44519, Egypt; gelmowalid@yahoo.com (G.E.); bactine_ho@yahoo.com (H.A.); ahhmht35@yahoo.com (A.A.H.A.)

**Keywords:** bovine viral diarrhea virus, cytopathic BVDV, immunosuppression, lymphocyte apoptosis, monocyte-derived macrophages, non-cytopathic BVDV

## Abstract

Bovine viral diarrhea virus (BVDV) is an important viral disease of cattle that causes immune dysfunction. Macrophages are the key cells for the initiation of the innate immunity and play an important role in viral pathogenesis. In this in vitro study, we studied the effect of the supernatant of BVDV-infected macrophage on immune dysfunction. We infected bovine monocyte-derived macrophages (MDM) with high or low virulence strains of BVDV. The supernatant recovered from BVDV-infected MDM was used to examine the functional activity and surface marker expression of normal macrophages as well as lymphocyte apoptosis. Supernatants from the highly virulent 1373-infected MDM reduced phagocytosis, bactericidal activity and downregulated MHC II and CD14 expression of macrophages. Supernatants from 1373-infected MDM induced apoptosis in MDBK cells, lymphocytes or BL-3 cells. By protein electrophoresis, several protein bands were unique for high-virulence, 1373-infected MDM supernatant. There was no significant difference in the apoptosis-related cytokine mRNA (IL-1beta, IL-6 and TNF-a) of infected MDM. These data suggest that BVDV has an indirect negative effect on macrophage functions that is strain-specific. Further studies are required to determine the identity and mechanism of action of these virulence factors present in the supernatant of the infected macrophages.

## 1. Introduction

Bovine viral diarrhea virus (BVDV) causes immune dysfunction in cattle and other ruminants, resulting in severe economic losses. The immune dysfunction associated with BVDV is believed to be, in part, a consequence of lymphoid depletion that could be mild or severe based on the virulence of the BVDV strains [1,2,3,4]. Furthermore, BVDV induces multiple abnormalities to immune cell function and immune mediators that could potentiate viral infection and pathogenesis [5,6]. BVDV infects neutrophils and decreases their microbicidal activity [7]. It also infects alveolar macrophage and decreases the expression of complement Fc receptors and chemokine production, reducing their ability to engulf opsonized pathogens [8].

Although only cytopathogenic (Cp) strains can induce direct apoptosis to peripheral blood mononuclear cells (PBMCs) [9,10], other studies have suggested that Ncp BVDV induces apoptosis in vitro in PBMCs isolated from BVDV-infected animals [11]. Ncp BVDV causes the most severe BVDV immune dysfunction and is associated with severe lymphoid depletion [2,3,12,13,14,15]. Some Ncp BVDV strains have been categorized under a new biotype, lymphocytopathogenic, as they can induce apoptosis in lymphocytes while not effecting epithelial cells [3,16]. In addition, Ncp BVDV has been reported to induce apoptosis in vitro in PBMCs isolated from BVDV-infected animals [11].

Factors associated with Cp BVDV-induced apoptosis include: (i) oxidative stress and (ii) soluble factor(s) produced by monocytes that induced apoptosis in uninfected cells after lipopolysaccharide (LPS) treatment, [9,10,17,18]. However, the exact mechanism by which Ncp/lymphocytopathogenic BVDV induced the apoptotic effect is not well understood. Several studies suggest the role of macrophages in lymphoid depletion and immune dysfunction associated with BVDV. These studies demonstrated the importance of monocytes/macrophage in lymphocyte apoptosis in a mixed population of PBMCs [9,13]. However, the role of apoptotic-associated cytokines, which are important for classic swine fever virus (CSFV)-induced lymphocyte apoptosis, do not appear to be important for BVDV [13]. BVDV and CSFV are classified under the same genus: *Pestivirus*, Family: *Flavivirdae.* It has also been hypothesized that the *Pestivirus*’ specific secreted Erns glycoprotein could play a role in the apoptosis mechanism associated with CSFV [19,20].

Immune mediators potentiate viral infection and pathogenesis. CSFV [21], and Cp BVDV [17], prime uninfected cells for activation-induced apoptosis. The mechanism for activation-induced apoptosis is virus-specific, where direct contact between infected and uninfected cells may also be involved in CSFV [21]. CSFV infection of bone marrow hematopoietic cells does not induce apoptosis in the majority of infected cells but does cause apoptosis in the uninfected cells, suggesting an apoptotic role for soluble factors released from infected cells. Cp BVDV induces apoptosis, while Ncp BVDV inhibits apoptosis [9,10,17,18,22,23].

Cp BVDV induces apoptosis in epithlial cellls in cell culture and in PBMCs, making it different from Lymphocytopathogenic BVDV, which can only induce apoptosis in lymphocytes. The soluble factor(s) involved in apoptosis has not been well characterized in any of these studies. Several studies have identified cytokines that prime for apoptosis, including IL-4 [24], TNF-α, IFN-γ [25,26,27] and IL1beta [28]. However, these cytokines are produced by lymphocytes rather than macrophages [23].

To investigate the indirect effect of different strains of BVDV on macrophage activity and lymphocytes apoptosis, both peripheral blood lymphocytes and a BVDV-free BL-3 cell line were used [3]. The direct apoptotic effect of BVDV was examined by the infection of lymphocytes with different BVDV strains. The indirect effect was investigated by infection of the MDM with one of two different strains of BVDV, followed by treatment of lymphocytes with the irradiated virus-free supernatant from BVDV-infected macrophages [29].

## 2. Materials and Methods

### 2.1. Viruses and Cells

Two Ncp strains of BVDV 2 were used in the current study: a high virulent 1373 and, a low virulent 28508-5, in addition to Cp 296C strain as control for apoptosis [30]. All strains were titrated using TCID50 titration assay [31].

BVDV-free Madin Darby bovine kidney MDBK cells (passage 113–135) were grown in minimal essential medium (MEM, Gibco BRL, Grand Island, NY, USA) (pH 7–7.4) supplemented with 10% BVDV-free fetal bovine serum (FBS) (Hyclone Laboratories, Logan, UT, USA), penicillin (100 U/mL) and streptomycin (100 µg/mL). MDBK cells were maintained at 37 °C with 5% CO_2_ in humidified incubator and were used for viral titration and propagation as well as apoptosis experiments.

Bovine lymphosarcoma cell line (Bl-3) is a non-adherent, BVDV-free B cell line clone derived from BL-3.1 cell line that was obtained from ATCC, Manassas, VA, USA. BL-3 cell line was cultured as previously described [3] and they were used to examine the direct and indirect apoptotic effect of different BVDV strains.

### 2.2. Isolation of Monocytes and Differentiation to Monocyte Derived Macrophage (MDM)

PBMCs were isolated as previously described [32,33] with modifications. Briefly, the heparinized blood was diluted 1:1 with PBS and overlaid over histopaque 1083 gradient using SepMate^TM^ 50 mL tubes (Stemcell Technologies, Cambridge, MA, USA) and centrifuged at 1200× *g* for 20 min at room temperature (RT). The buffy-coat was then transferred to a clean 50 mL conical tube and washed five times with PBS, followed by centrifugation at 120× *g* for 10 min at RT. The viability of PBMCs was determined by trypan blue exclusion assay according to Strobber [34]. The PBMCs were suspended in RPMI 1640 medium (GE Healthcare, Hyclone Laboratories, Logan, UT, USA) supplemented with 10% FBS, penicillin (100 U/mL) and streptomycin (100 µg/mL) to achieve a final concentration of 1 × 10^6^ cells/mL. The cells were incubated in T175 flasks for 3 h at 37 °C. Then, the adherent monocyte was washed with PBS five times and detached by incubation for 10 min with Accutase^TM^ (eBioscience, San Diego, CA, USA). Detached monocytes were PBS-washed twice to get rid of Accutase. The isolated monocyte cultured in complete RPMI 1640, as described by Elmowalid [29], at a concentration of 10^5^ cells/well in 48-well plate, followed by incubation for 5 days at 37 °C. The incubated cells were fed every other day by replacing half of the conditioned media with fresh complete RPMI. At day 5, the MDM were characterized phenotypically as MHCI-, MHCII-, CD11b- and CD14-positive cells.

### 2.3. Production and Inactivation of Infected MDM Supernatant

The MDM were infected with 100 µL of 10^5^ TCID50 BVDV strains at MOI of 1 in triplicate as described by Elmowalid [29], with modifications. The infected cells were incubated for 1 h, then washed to remove the excess unbound virus, and 500 µL of complete RPMI 1640 medium was added to each well in a 24-well plate. At least one column of the plate was mock-infected with complete RPMI 1640 medium as a negative control. The infected MDM were incubated for 12, 24 or 48 h at 37 °C in CO_2_ incubator. The BVDV-infected macrophage supernatants were collected at 12, 24 and 48 hpi (hours post-infection) and centrifuged at 1000× *g* for 10 min at RT to remove cellular debris. The supernatant was UV-inactivated for 20 min on ice to exclude the direct virus effect [35]. The absence of any infectious viral particles in the treated supernatants was confirmed by inoculation on MDBK cells followed by a 5-day incubation and BVDV specific immune-staining of the inoculated MDBK cells using both immune-peroxidase and immunofluorescence. The positive control was 1373 infected MDBK cells.

### 2.4. Phagocytosis

One mL of virus-free (UV-inactivated) supernatants collected at 24 or 48 hpi from Ncp1373 or 28508-5 BVDV strains or mock-infected MDM were used to treat MDM, cultured on 24-well plates (approximately 5 × 10^5^ cells/well whose viability was >92%), for 24 h. This was followed by washing the treated cells twice with PBS and then exposed to 250 µL containing approximately 2 × 10^7^ of TRITC-labeled *C. albicans* (50 yeast/macrophage) and incubated for 30 min at 37 °C. Finally, the cells were washed twice in cold PBS and re-suspended in 200 µL/well freshly prepared paraformaldehyde (PFA) to be examined under UV-microscopy. The number of yeast/cell counted and the cells were classified into two groups: cells that contained >20 TRITC labeled yeast/cell that indicated normal phagocytic activity and cells that contained <20 TRITC labeled yeast/cell that indicated insufficient phagocytic activity. A total number of 100 MDM containing yeast were counted and the percentage of phagocytic activity was calculated according to the following formula: Phagocytic% = (number of MDM containing >20 yeasts/total number of MDM) × 100.

### 2.5. Bactericidal Activity

MDM were treated with 1 mL/well of UV-inactivated supernatant from MDM infected with 1373, 28508-5 or mock-infected 24 and 48 h supernatant, and incubated for 24 h at 37 °C. The cells were then washed twice in PBS and incubated for 30 min or more (up to 120 min) with 1 × 10^7^
*E. coli* (20 bacteria/macrophage) suspended in RPMI 1640 supplemented with 5% FBS and no antibiotics to allow bacterial phagocytosis. One group of the cells were washed three times with cold PBS 30 post-incubation to stop phagocytosis and to remove excess bacteria (time 0) and were lysed and the lysates were centrifuged at 800× *g* for 10 min and the supernatants were aspirated, while the pellets were suspended in LB broth and stored at 4 °C. The other group of cells were incubated in 5% FBS, and no antibiotics for extra 90 min to allow killing (total 120 min) followed by washing and lysis, as described above. Cell lysates of both 0 and 90 min timepoints were placed in 96-well plates and incubated for 18 h at 37 °C, followed by adding 30 µL/well of of 50 mg/mL 3-(4,5-dimethylthiazol-2-yl)-2,5-diphenyl tetrazolium bromide (MTT, Sigma, St. Louis, MO, USA) and the plates were incubated at 37 °C for another 6–8 h. Optical density was measured at 540 nm wavelengths in ELISA reader (Biotek, ELx808, Winooski, VT, USA). The percentage of killing was calculated by the following formula: Killing% = (absorbance after incubation for 90 min/Absorbance at time 0) × 100.

### 2.6. Nitric Oxide Production

Nitric oxide (NO) assay was done in 24-well plates. MDM were treated by adding 1 mL/well of UV-treated supernatant from 1373 or by direct infection with 1373 strain of BVDV (MOI of 1) kept with the cells. Mock-infected supernatant or RPMI 1640 media containing 5% FBS were used as a control. The cells were incubated for 24 h at 37 °C. The cells were then stimulated with lipopolysaccharide (25 µg/mL) (LPS, Sigma, St. Louis, MO, USA) for another 24 h. Then, supernatants were collected and nitric oxide (NO) concentration was determined using Griess reagent (Sigma, St. Louis, MO, USA) using Nitrite (NO_2_) measurement as an indicator of NO production according to the protocol of Blond et al., 2000: briefly, 100 µL of culture supernatant was added to 100 µL of Griess reagent, made of a 1/1 mixture of 1% (wt/vol) sulfanilamide and 0.5% (wt/vol) *N*-(1-naphthyl)ethylenediamine dihydrochloride (Sigma, St. Louis, MO, USA) in 30% acetic acid, in each well of a 96-well plate. Reactions were performed in triplicate at RT for 10 min. Chromophore absorbance was then measured at 550 nm in a microplate reader (Bio-TEK Instruments model EL 311; OSI, Paris, France). Nitrite concentration was evaluated by comparison with the nitrate standard curve (Sigma, St. Louis, MO, USA). The lower limit of detection for nitrite is 250 nM.

### 2.7. Immunostaining and Flow Cytometry of CD14 and MHC II

To examine the effect of BVDV direct infection or its supernatant on macrophage surface marker expression, the MDM were either infected or treated with different BVDV-48 h-supernatants: including 1373 and 28508-5 strains. This was in addition to the supernatant from mock-infected MDM. Surface marker expression was investigated 48 hpi, starting with using Accutase^TM^ (eBioscience, San Diego, CA, USA) to detach the MDMs that were washed twice in PBS and plated in a U-bottom-96-well plate by adding100 µL of cells (2 × 10^5^ cell) per well. Cells were then suspended in blocking buffer (50 µL/well) (PBS containing 2% FCS and 0.1% sodium azide) for 20 min at RT. 50 µL/well of mouse anti-bovine CD14, or MHC II (VMRD, Pullman, WA, USA) diluted at 1:200 in PBS was added and the cells incubated for 20 min at 4 °C, followed by washing three times in PBS. Subsequently, the cells were incubated with FITC-conjugated goat anti-mouse Ig G (Ig G whole molecule, ICN/Cappel laboratories, Santa Mesa, CA, USA) diluted at 1:1000 in staining buffer for 20 min in dark at 4 °C, followed by washing three times in PBS. Fifty (50) µL/well of propidium iodide (1 µg/mL) were added directly before analysis to stain the dead cells. Both cell control (without staining) and the FITC control (just FITC staining without the primary anti-bovine antibodies) were included to eliminate nonspecific background. For each sample, mean fluorescent intensity of immune-stained cells was estimated and analyzed on FACScan flow cytometry (Becton Dickinson, Mountain View, CA, USA) using CELL Quest software (BD Biosciences).

### 2.8. Examining the Indirect or Direct Effect of BVDV Strains on Peripheral Blood Lymphocyte Population and MDBK

To examine the indirect effect of BVDV strains, peripheral blood lymphocytes, BL-3 or MDBK were treated with various time-point irradiated supernatants from BVDV-infected macrophages at a dilution of 1:1 and incubated for 12, 24 or 48 h post treatment, as previously described [36]. To examine the direct effect of BVDV strains, peripheral blood lymphocytes or BL3 were infected with BVDV strain at MOI of 1, as described by Ridpath et al. [37]. The effect of the direct infection of lymphocyte with Cp 296C strain was included as a positive apoptosis control.

### 2.9. Chromatin Condensation

The apoptotic effect of the infected MDM supernatant on epithelial cells was investigated by incubating MDBK cells for 12, 24, or 48 h at 37 °C with 1 mL/well of UV-treated supernatant from 1373, or 28508-5 as well as the non-infected supernatant. The cells were then washed twice in PBS, and then stained using 4,6-diamidino-2-phenylindole (DAPI, Sigma, St Louis, MO, USA) according to the protocol of Mi Hyeon et al., 2002. Simply, one volume of pre-staining solution [citric acid (2.1 g), Tween 20 (0.5 mL), and distilled H_2_O (100 mL)] was added and the cells were incubated for 5–7 min at RT in the dark. After the incubation, another six volumes of staining solution [(citric acid (2.1 g), DAPI (0.2 mg/mL), 1 mg DNAse, and distilled H_2_O (100 mL)] were added for another 5–7 min. Cells were washed three times in PBS and examined under fluorescence microscopy. Two hundred cells (200) were counted and the mean value was calculated. The apoptotic index was the mean of three independent experiments.

### 2.10. Annexin V Staining

Both direct BVDV infection or the indirect effect of its supernatant on bovine lymphocytes and MDBK cells was investigated using the AnnexinV staining kit (eBioscience, San Diego, CA, USA). Briefly the infected or supernatant-treated lymphocytes were washed once with PBS and the cells were re-suspended in 100 µL of 1x binding buffer supplied by the kit. Five µL of FITC conjugated annexin v antibody added to the lymphocytes suspended in 100 µL of binding buffer and incubated for 10 min in the dark. This was followed by washing the cells twice by suspending the cells in 1x binding buffer followed by adding another five µL of propidium iodide to exclude the dead cells. Finally, the cells were re suspended in 200 µL of 1x BD FACSTM lysing solution followed by analysis with BD Accuri^TM^ C6 Plus Flow Cytometer (BD Biosciences, San Jose CA, USA). The result was expressed as % of apoptosis.

### 2.11. Protein Electrophoresis

The BVDV-infected MDM supernatant was fractionated on 10% sodium dodecyl sulphate polyacrylamide gel (SDS-PAGE) according to the protocol of Schagger and von Jagow, 1987. Briefly, 40 µL of the supernatant (around 50 µg) were added to 10 µL of 5x sample buffer (65 mM Tris-HCL, pH 7.0, 2% SDS, 10% glycerol, 5% β mercaptoethanol, and 0.001% bromophenol blue) with heating at 100 °C for 3–5 min. Protein electrophoresis was then done by loading the treated sample (50 µL) on the SDS-PAGE gel at 25 mA current for 45–60 min followed by staining with Coomassie blue overnight (in some experiments for 4 h) with shaking at RT. Finally, the gel was immersed in de-staining buffer (40% methanol and 10% glacial acetic acid) with shaking for 15 min at RT until the protein bands become clear [38].

### 2.12. Quantification of Apoptosis-Related Cytokines by qRT-PCR of BVDV-Infected MDMs

Both the infected and mock-infected MDM were pelleted at 500× *g* for 8 min, then washed twice in PBS. The infected MDM were lysed and nucleic acid (NA) extraction done using RNeasy extraction kit (Qiagen, Valencia, CA, USA). The extracted NA of different samples was normalized to 5 ng/µL using Nanodrop ND-1000 Spectrophotometer (Fisher Scientific, Portsmouth, NH, USA). Nucleic acid extracted from mock-infected MDM with complete RPMI 1640 medium was used as a negative control while NA extracted from MDM treated with Concanavalin A (ConA) or lipopolysaccharide (LPS) (Sigma-Aldrich, St. Louis, MO, USA) was used as a positive control. Around 5 µL of the normalized NA was used for relative quantification of apoptosis-related cytokines; TNF-α, IL-1α, IL-1β and IL-6 for each sample in duplicates using quantitative reverse transcriptase PCR (qRT-PCR) and Power SYBR^®^ Green RNA-to-CtTM 1-Step Kit (Thermo Fisher Scientific, Millersburg PA, USA). Additional cytokines were also measured that include: IFN–α, β and γ, IL-4, 8, 10, and 12. The cytokine primers used were described in Table 1 [39]. The relative expression of mRNA was standardized using beta actin and GAPDH as housekeeping genes. Each qPCR experiment was followed by heat dissociation curve step to exclude nonspecific amplification. qRT-PCR results were analyzed using relative expression software tool (REST©2009 software) [40].

### 2.13. Supernatant Neutralization

To exclude the apoptotic effect induced due to possible viral secreted proteins, 24 or 48 h supernatants from the BVDV-infected macrophages with 1373 strain were incubated at 37 °C for 1 h with Erns specific 15c5 monoclonal antibody (IDEXX Laboratories, Westbrook, ME, USA) or with BVDV-polyclonal antibody (kindly supplied by Dr. Robert Fulton, Food Animal Research, Oklahoma State University). The mixture of the infected supernatant and antibody was prepared as two parts of infected supernatant incubated with one part of concentrated antibody. Three (3) parts of the supernatant/antibody mixture were added to one part of BL-3 cell line seeded at 1 × 10^5^ cells/well, followed by incubation at 37 °C for 36 h post-treatment.

### 2.14. Statistical Analysis

Data were analyzed using a Student’s *t*-test (Microsoft EXCEL, MAC 2011) to assess the significance of the differences between mean values of treated and control samples at each timepoint. Differences were considered significant at *p* < 0.05, however some treatments showed very significant difference with *p* value of <0.01. Every experiment was achieved using at least three different animals and each experiment was done in triplicates. The variations in results were calculated by standard deviations at each timepoint. For cytokine analysis, REST© 2009 [40], that is based on analysis of variance (AVOVA).

## 3. Results

### 3.1. Inactivation of Infected Supernatant

The irradiated supernant-treated MDBK cells had no immune-peroxidase stain for the UV-irradiated supernant (Figure 1A) compared to the positive intracytoplasmic staining of 1373-infected-MDBK non-irradiated control (Figure 1B). There was no fluorescence signal for the irradiated strain (Figure 1C) compared to the non-irradiated infection control (Figure 1D).

### 3.2. Effect of Supernatant from BVDV Infected MDM on Macrophage Phagocytic Activity Compared to the Direct BVDV Effect

At 24 h post-treatment, the supernatant from Ncp 1373 significantly reduced MDM phagocytic activity by 38.5%, which continued to decrease to 51% at 48 h post-treatment compared to the supernatant from Ncp 28508-5, which was similar to the mock-infected control at any timepoint (Figure 2). Direct infection of MDM with the highly virulent 1373 strain, but not 28508-5, diminished phagocytic activity of MDM as early as 12 h post-treatment by 27.5% (Figure 2). We also found that heat-treated supernatant from 1373, or 28508-5 BVDV-infected supernatant had no effect on MDM phagocytic ability (data not shown).

### 3.3. Effect of Supernatant from BVDV-Infected MDM on Macrophage Bactericidal Activity Compared to the Direct BVDV Effect

No significant effect was observed at 6 or 12 h post-treatment, however, the supernatant from Ncp 1373 strain significantly (*p* < 0.05) reduced MDM bactericidal activity by 37.7% and 51.4% at 24 and 48 h post-treatment, respectively (Figure 3). Direct infection of MDM with 1373 decreased the bactericidal activity by 22%, which began as early as 12 hpi, unlike the late indirect effect. The inhibition of bactericidal activity was time-dependent, and continued to increase to reach 54.7% by 48 hpi. There was no significant effect with 28508-5 strain, its corresponding infected-MDM supernatant, mock-infected, or the non-infected supernatant on bactericidal activity (Figure 3).

### 3.4. Effect of Supernatant from BVDV Infected MDM on Macrophage Nitric Oxide Production Compared to the Direct BVDV Effect

The direct infection of MDM with both BVDV strains significantly increased NO production (158–164 µmol/mL), while neither 1373- nor 28508-5-infected MDM supernatant affected NO production in uninfected MDM (Figure 4). Both supernatant treatments, as well as non-infected supernatant and mock-infected controls, showed low nitric oxide production (93.3 µmol/mL) (Figure 4).

### 3.5. Effect of Supernatant on MHC II Expression Compared to the Direct BVDV Effect

At 24 h post-treatment, cells treated with 1373 supernatant had significant reduction in MHC II expression to 61.0% (Figure 5A). At 48 h, MHC II expression was further reduced to 50.0% (Figure 5A) In contrast, 28508-5 or mock-infected supernatant had no significant reduction in MHC II expression and the expression level was greater than 94.0% at 24 or 48 h (Figure 5A).

### 3.6. Effect of Supernatant on CD14 Expression Compared to the Direct BVDV Effect

At 24 h post-treatment, the CD14 expression decreased in MDM infected with Ncp 1373 strain or it’s supernatant (Figure 5B). At 36 h post-treatment, CD14 expression further decreased to 44.6% and 49.7% for Ncp 1373 and its supernatant, respectively (Figure 5B). The heat-treated supernatant or supernatant of 28508-5 did not show a significant decrease in the percentage of cells expressing CD14 and the average percentage was over 93% at 24 or 36 h (Figure 5B).

### 3.7. Effect of Supernatant from BVDV Infected Macrophages on MDBK Cells

Chromatin condensation was detected after 24 h in treated MDBK with 1373 supernatant, while no chromatin condensation was detected in MDBK treated with 28508-5 nor non-infected supernatant (Figure 6A). The percentage of apoptotic cells was not significant until 24 h post-infection using Annexin V assay. It increased from 36.3% (24 h post-treatment) to reach 51.3% (48 h post-treatment) in contrast to less than 3% in case of mock-infected or treated MDBK with either 28508-5 or non-infected supernatant (Figure 6B).

### 3.8. Effect of Supernatant from BVDV-Infected Macrophages on Different Lymphocyte Populations

The 12 h supernatants did not induce apoptosis compared to the mock-infected control (Figure 7). Both 24 and 48 h supernatants of the highly virulent 1373 resulted in an apoptotic effect on either freshly isolated (data not shown) or BL-3 lymphocytes. All supernatants of the low virulent Ncp 28508-5 at different timepoints did not induce apoptotic changes compared to the mock-infected control for either BL-3 lymphocytes or peripheral blood total lymphocyte population (Figure 7A,B).

### 3.9. Effect of Direct Infection of Lymphocyte with BVDV Strains

Only the Cp 296C strain was able to induce significant lymphocyte apoptosis as early as 12 h post-infection compared to the non-infected/mock control and the Ncp BVDV strains (Figure 8A) The direct infection of lymphocytes with 1373 was compared to the indirect effect of this virulent BVDV strain. Interestingly, the direct infection of lymphocytes did not induce significant changes compared to the mock-infected control until 48 h post-infection with 21.8% as the maximum apoptotic effect. The indirect effect of 1373 was tow times greater than the direct infection of lymphocyte at 24 and 48 hpi, with a maximum apoptotic effect of 35% at 48 h post-treatment (Figure 8B).

### 3.10. Supernatant Protein Analysis

No viral proteins were detected in the supernatants at 6 hpi (Figure 9A), however, faint bands of 40–45 KD started to appear in the 1373-infected MDM supernatant at 12 hpi (Figure 9B). By 24 hpi, bands of 30 and 120 KD were observed in supernatant from Ncp1373. MDM culture infected with 28508-5 showed only one dense band of about 40–45 KD (Figure 9C). At 48 hpi, the band numbers and densities increased in MDM infected with the Ncp 1373. In contrast, only two bands of 40 to 45 KD were observed in supernatant from 28508-5 Ncp BVDV strains by that time (Figure 9D). No bands were observed in supernatant from mock-infected MDM at 6, or 12 h. At 24 or 48 hpi, only one band of 40–45 KD was observed in the mock-infected cells supernatant (Figure 9).

### 3.11. Role of Cytokines in the Indirect Lymphocyte Apoptosis

Apoptosis-related cytokines including TNF-α, IL-1β and IL-6, did not show significant change at any of the three timepoints compared to mock-infected control. Other cytokines like IFN-γ, IL-10, IFN-α, IFN-β, IL-12, IL-4, IL-8 and TGF-β have also shown no significant difference between groups. The positive control mixture of Concavalin A and LPS upregulated all of the cytokines (data not shown).

### 3.12. Role of Viral Factors in the Indirect Lymphocyte Apoptosis

There was no significant difference in the percentage of lymphocyte apoptosis between the neutralized supernatant with either mAb or polyclonal Ab and the un-neutralized supernatant (data not shown).

## 4. Discussion

This study sheds light on the role of macrophages in the pathogenesis of BVDV infection. Phagocytosis is an important immune response of the macrophage and constitutes one of the first defenses against microbial invasion. Successful phagocytic activity leads to the initiation of successful immune response. In this study, supernatant from the highly virulent 1373 strain reduced macrophage phagocytic activity. In a previous study, Ncp BVDV infection of alveolar macrophage decreased expression of complement and Fc receptors and chemokine production, reducing their ability to engulf opsonized pathogens [8]. IL-2 inhibitory substances are produced in BVDV-infected PBMCs cultures that bind to receptors on lymphocytes, macrophages and dendritic cells that activate protein kinase C, resulting in the phosphorylation of proteins and inhibition of basic metabolic activities [41], that may reduce cell movement and the capacity to engulf organisms. A reduction in macrophage phagocytic capacity may be due to the presence of chemokine inhibitory mediators or the decrease in cytokine-induced chemotaxis [42] that lead to the decreased capacity of phagocytic cells for chemotaxis and phagocytosis.

Microbicidal activities are one of the main functions of macrophage to control and clear microbial infection. In vivo experiments suggest that BVDV immunosuppression resulted in subsequent secondary microbial infection. The supernatant from low virulent 28508-5-infected macrophage had no effect on bactericidal activity, while the highly virulent 1373 supernatant decreased the bactericidal activity of macrophages significantly with more than 50% by 48 hpi (*p* < 0.05). Considerable evidence has accumulated that BVDV may be a key component in multiple-etiology diseases [43].

Nitric oxide production was not affected by the supernatant exposure. Direct infection of MDM with highly virulent 1373 strain significantly induced NO production; however, 1373-infected macrophage supernatant (indirect) had no significant effect on NO production in MDM following stimulation with LPS. Another flavivirus, the hepatitis C virus (HCV) ERNS protein and HIV infection was associated with increased NO synthase (NOi) production in mice livers [44,45]. Type I IFN is a potent inducer of NO; however, it is downregulated by Ncp BVDV strains [25,46]. Taking these results together, Ncp BVDV might induce NO production directly through ERNS protein activation to NOi, but not indirectly due to the lack of secreted IFN in the MDM supernatant.

Microbial recognition, processing and presentation to lymphocytes is one of the unique functions of antigen-presenting cells for initiation of immune response. The recognition of pathogens is achieved through surface markers, which initiate the immune response. Our results indicated a significant reduction in surface marker expression of MHC II and CD14, associated with the highly virulent 1373-infected MDM supernatant. MHC II is an important marker for the initiation of immune response and antigen presentation. The current study indicated that the highly virulent 1373 reduced the surface expression of MHC II molecules. A previous study demonstrated that MHC II expression decreased in PBMCs infected with Ncp BVDV [47]. Another study reported that BVDV infection downregulated MHC I and II in monocytes [48]. This is consistent with what we found but in macrophage.

CD14 is important for toll-like receptors and gram-negative recognition and phagocytosis. Our findings indicated that low virulent 28508-5 supernatant had no effect on CD14 expression, while the highly virulent 1373 supernatant significantly decreased CD14 expression. CD14 downregulation was associated with the reduction in the phagocytic activity of neutrophils in HCV patients [49]. In human and mouse models, downregulation of CD14 was associated with suppression of antigen-specific lymphocyte proliferation [50]. Impairment of macrophage inflammatory function is a likely consequence of this surface reduction with subsequent secondary microbial invasion and the immunosuppressive effect of BVDV on the infected animals.

The findings of this study indicate the importance of soluble factors induced in BVDV-infected macrophages in viral pathogenesis. These results demonstrate that only the highly virulent Ncp BVDV strain induced the production of soluble factors capable of impairing macrophage inflammatory functions and surface marker expression as early as 12 h post-infection. These mediators and factors also induced epithelial cell and lymphocyte apoptosis. The release of several BVDV induced-soluble factors release has been reported [8,51,52,53]. On gel electrophoresis, several proteins ranging from 30 to 120 kDa were produced in 1373-infected MDM supernatants that were either less intense or not present at all in 28508-5-infected MDM supernatant. In a previous study, inflammatory mediators of 40–75 KD play a pivotal role in viral infection and pathogenesis [54].

Another study indicated that PMNC infected with Ncp or Cp released unknown factor(s) in the culture media that could be of cellular or viral origin [52,55]. Previously, secreted Erns viral glycoprotein played an important role in lymphocyte apoptosis induced by BVDV and CSFV [19]. In our study, we found no association of Erns glycoprotein or any other viral factors in the induction of apoptosis. Neither anti-ERNS mAb or a polyclonal BVDV Ab treatment of BVDV-infected MDM supernantant reduced lymphocyte apoptosis. The difference between BVDV and CSFV may be due to the use of a cloned Erns glycoprotein of CSFV that would be expressed at higher levels versus constitutive expression of BVDV Erns in BVDV-infected cells.

In the current study, we found that a supernatant from the highly virulent 1373 infected MDM induced apoptosis in MDBK as early as 24 h post-treatment, while low virulent 28508-5 infected MDM supernatant did not induce apoptosis. Previous work has shown that Ncp 28508-5 BVDV strain did not cause an inflammatory response in vivo, while Ncp 1373 caused inflammation and depletion of lymphocytes [56]. Interestingly, the Ncp 28508-5 strain is much more successful at establishing persistent infection. This lack of an inflammatory response may be one evasion mechanism to avoid the immune response and establish persistent infection in the developing fetus.

Furthermore, we found that the virulence of the BVDV strain is very important in determining the severity of the corresponding lymphoid depletion and subsequent immune suppression. The low virulence of the 28508-5 supernatant failed to induce apoptosis on the freshly isolated lymphocytes, while the highly virulent 1373 supernatant induced significant lymphocyte and somatic cell apoptosis. These results support the association between virulence and the degree of immune suppression that is associated with lymphoid depletion in vivo [1,2,3,57].

The role of macrophages in lymphocyte apoptosis was investigated in this study. Our results suggested that the supernatant of infected macrophages with only virulent BVDV strains induced lymphocyte apoptosis. These results provide an explanation for the observation that there is an increased number of macrophages in lymph nodes 3 days post-infection prior to lymphoid depletion [13]. Moreover, this indirect effect mediated by macrophages was not significant until 24 h post-infection with virulent BVDV strains, suggesting that rapid treatment strategies may help to limit the infection and would decrease or prevent the severe immune suppression induced by BVDV.

Our data support the results of Pedrera et al., 2009 [13] that did not relate the apoptotic effect to the biotype of the strain but to its virulence. Interestingly, we noted that the highly virulent 1373 strain of BVDV induced a more marked and faster apoptotic effect on lymphocytes indirectly as compared to the direct effect. These results suggest that there is a more severe and rapid immune suppression associated with the highly virulent 1373 that is likely due to the indirect effect mediated by macrophages.

Apoptosis-related cytokines produced by BVDV-infected macrophages did not play a role in lymphocyte apoptosis. There was no significant change in TNF-α, IL-1α, IL-1β or IL-6 transcriptional levels. These results were different from those described for CSFV, where TNF-a was involved in the indirect lymphocyte apoptosis activated by the virus [36]. Although TNF-α induction appears to be included in the mechanism of CSFV-induced lymphocyte apoptosis, this does not appear to be true for BVDV. The lack of TNF-α induced apoptosis is consistent with previous research with BVDV lymphocyte apoptosis [13]. Another study suggested that a role of IFN-γ in inducing apoptosis, unlike our research, was done using monocytes and a different BVDV strain [26].

The BL-3 cell line gave similar results, as seen with the peripheral blood lymphocytes. BL-3 cell findings were consistent with previously reported apoptotic effect of the highly virulent BVDV strains on the BL-3 cell line [30]. The degree of apoptosis induced in the lymphocytes of both peripheral blood or BL-3 cell line was similar. This finding leads us to speculate that B cells are more susceptible than T lymphocytes, but this hypothesis needs to be tested.

Another study suggested the identity of some possible factors that mediate the lymphocyte apoptosis post-BVDV infection, like micro RNA and programmed death-1 (PD-1) [58,59]. Finally, the identification and characterization of these factors and the molecular mechanisms involved in their induction and modes of action will be important goals for future studies.

## 5. Conclusions

BVDV causes immune dysfunction that leads to vaccination failure and secondary bacterial infection with lymphoid depletion associated with BVDV clinical cases. It was important for our study to focus on immune dysfunction caused by BVDV through investigating the indirect effect of an infected macrophage supernatant of various BVDV strains on macrophage inflammatory function as well as lymphocyte apoptosis, these two cells that are important for clearance of the virus from the body of infected animals. Our data suggested that the immune dysfunction associated with the highly virulent Ncp 1373 strain was mainly due to the indirect effect mediated by macrophage-secreted mediators. These mediators may impair macrophage surface marker expression, which in turn disrupt the general macrophage inflammatory function, including phagocytic and bactericidal activity but not NO production. These soluble factors also enhance lymphocyte apoptosis with subsequent lymphoid depletion associated with BVDV cases. Unlike CSFV, neither TNF-a nor secreted Erns glycoprotein induces BVDV lymphocyte apoptosis. This led us to the hypothesis that host but not virus factors likely mediated the indirect lymphocyte apoptotic effect.

Further investigations of the factors present in the supernatant of the infected macrophages with the highly virulent strains need to be conducted so that we have a better understanding of the disease mechanism. This can lead to finding a way to block these factors that can be used as a therapeutic practice in acutely infected animals to help improve recovery. Understanding these factors better would also help to improve the current control strategies and decrease the risk and the economic losses of BVDV. Future studies that include other BVDV strains need to be conducted to further confirm our findings.

## Figures and Tables

**Figure 1 viruses-12-00701-f001:**
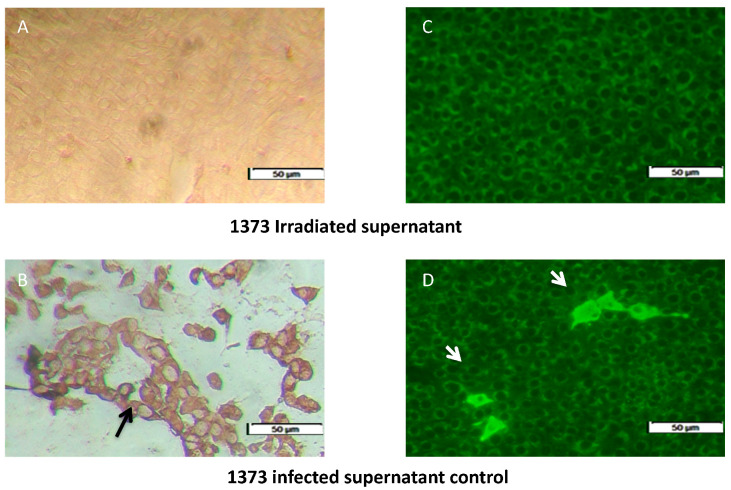
Successful inactivation of 1373-infected monocyte-derived macrophages (MDM) supernatant by UV-irradiation. (**A**,**C**) irradiated 1373-supernatant that show no evidence of intra-cytoplasmic replication by immune-peroxidase (IP) and immune-fluorescence (IF). (**B**,**D**) intra-cytoplasmic replication signal of the infected MDM supernatant, indicated by black and white arrows in the IP and IF assay, respectively (**B**,**D**).

**Figure 2 viruses-12-00701-f002:**
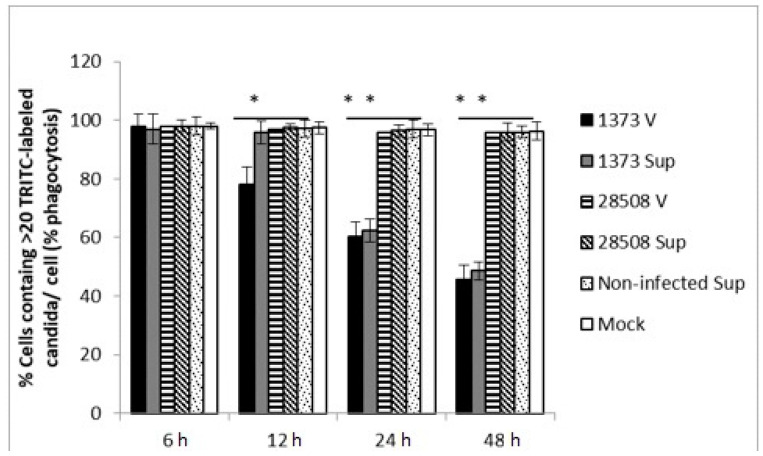
The effect of BVDV strains and the corresponding infected-MDM supernatants on macrophage phagocytic activity. Phagocytic index is defined as the % of MDMs that contains >20 TRITC-labeled *C. albicans*. h: hours post-treatment, V: direct virus infection, no supernatant, Sup: BVDV-infected macrophage supernatant, non-infected Sup: non-infected macrophage supernantant, Mock: culture medium *: *p* < 0.05 (>95% confidence), **: *p* < 0.01 (>99% confidence).

**Figure 3 viruses-12-00701-f003:**
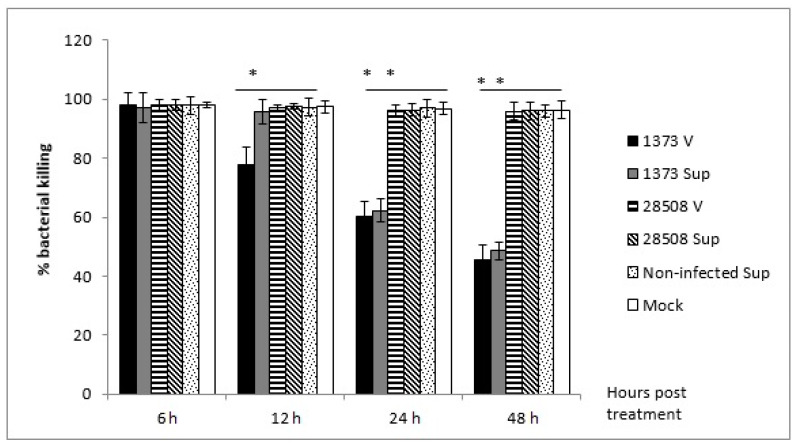
The effect of BVDV strains and the corresponding infected-MDM supernatants on macrophage bactericidal activity for 48 h post-treatment. Sup: supernatant (indirect effect), V: virus (direct effect). *: *p* < 0.05 (>95% confidence), **: *p* < 0.01 (>99% confidence).

**Figure 4 viruses-12-00701-f004:**
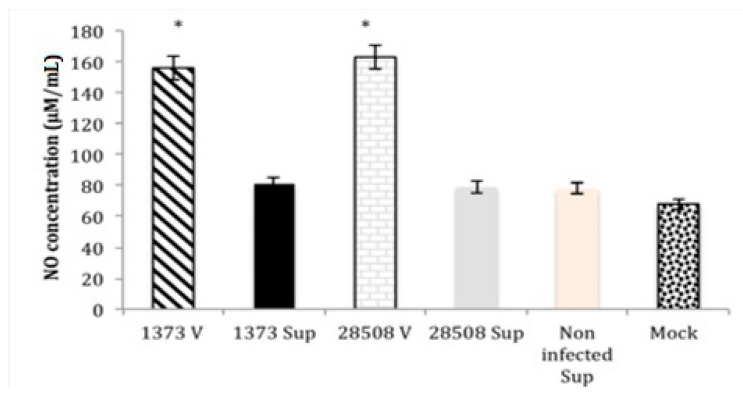
Comparison between the direct effect of BVDV strains and the corresponding infected-MDM supernatants (the indirect effect) on macrophage nitric acid production 24 h post-treatment. Sup: supernatant (indirect effect), V: virus (direct effect). *: *p* < 0.05 (>95% confidence).

**Figure 5 viruses-12-00701-f005:**
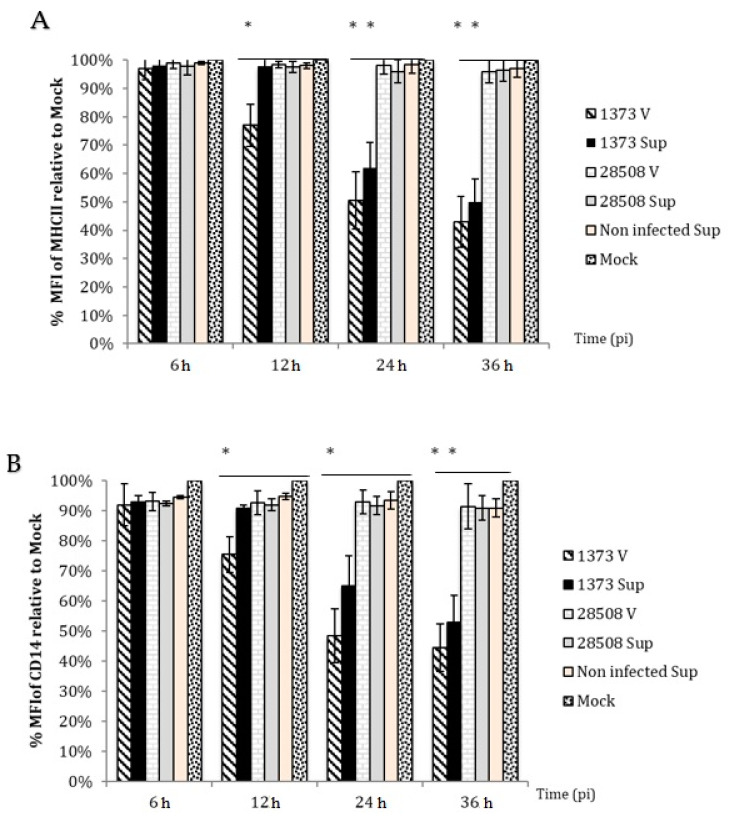
The effect of BVDV strains and the corresponding infected-MDM supernatants (indirect effect) on surface marker expression. (**A**) MHC II and (**B**) CD14 surface marker expression. Sup: supernatant (indirect effect), V: virus (direct effect). *: *p* < 0.05 (>95% confidence), **: *p* < 0.01 (>99% confidence).

**Figure 6 viruses-12-00701-f006:**
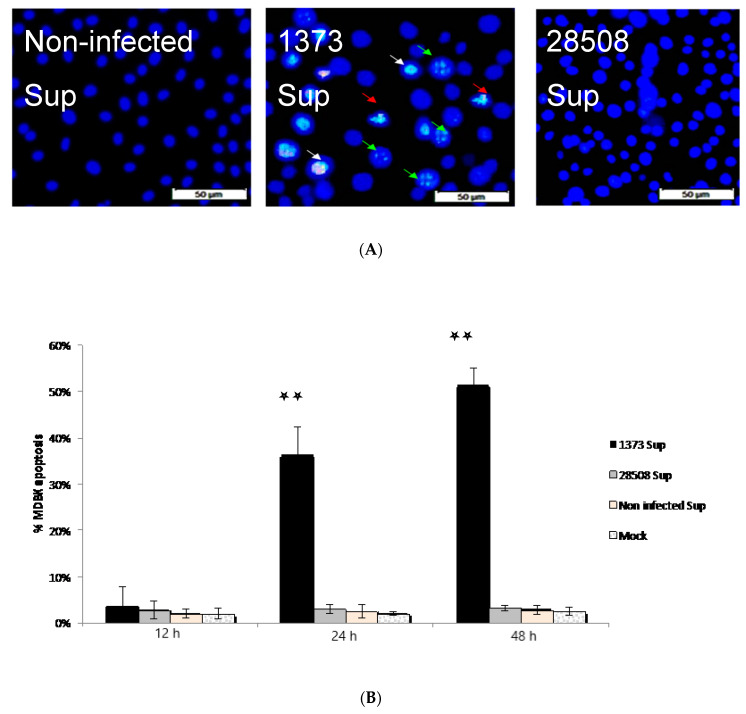
The indirect (infected MDM 24 h supernatant) effect of different BVDV strains on MDBK apoptosis. (**A**) Chromatin condensation by 1373 strain in contrast to 28508-5 and non-infected supernatant. (**B)** The % of MDBK apoptosis in response to different supernatants. 1373 Sup: BVDV 1373-infected macrophage supernatant, 28508-5 Sup: BVDV 28508-5-infected macrophage supernatant Non-infected Sup: non-infected macrophage supernantant, Mock: culture medium **: *p* < 0.01 (>99% confidence).

**Figure 7 viruses-12-00701-f007:**
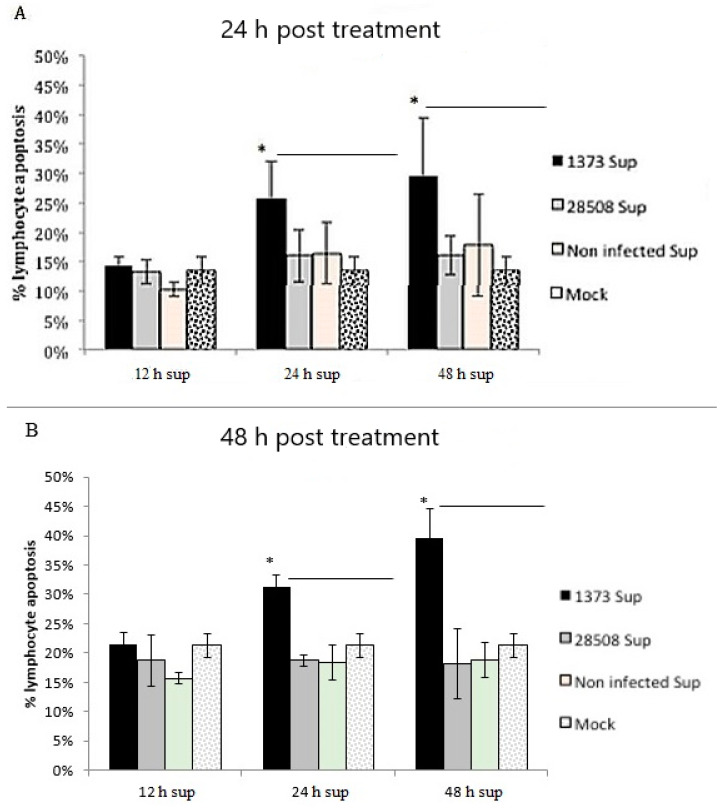
The indirect (infected MDM 12, 24 and 48 h supernatant) and the direct effect of different strains of BVDV on lymphocyte apoptosis. (**A**) The indirect effect 24 h post-treatment (**B**) The indirect effect 48 h post-treatment 1373 Sup: BVDV 1373- infected macrophage supernatant, 28508-5 Sup: BVDV 28508-5-infected macrophage supernatant non-infected Sup: non-infected macrophage supernatant, Mock: culture medium, *: *p* < 0.05 (>95% confidence).

**Figure 8 viruses-12-00701-f008:**
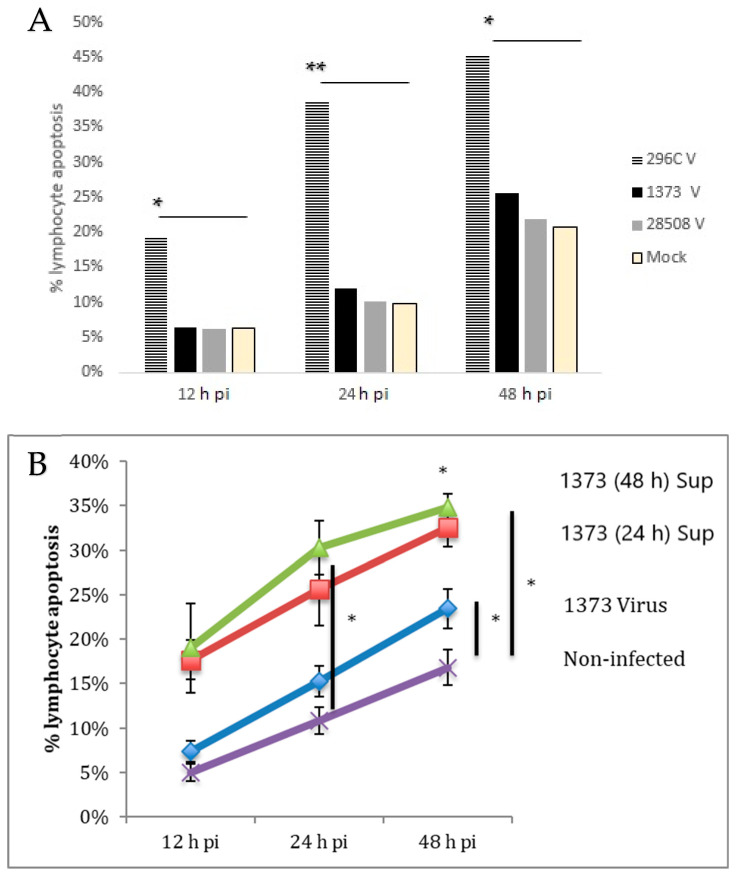
(**A**) The direct effect of BVDV strains on lymphocyte apoptosis (**B**) Comparison between the direct infection and the indirect (infected MDM 24 and 48 h supernatant) effect of the highly virulent 1373 BVDV strain on lymphocyte apoptosis for up to 48 pi. 296C V: direct lymphocyte infection with Cp 296C strain, 1373 V: direct lymphocyte infection with Ncp 1373 strain, 28,508 V: direct lymphocyte apoptosis with Ncp 28508-5 strain, Mock: non-infected lymphocyte, 1373 (48 h) Sup: supernatant from 1373-infected macrophage at 48 h post-treatment, 1373 (24 h) Sup: supernatant from 1373-infected macrophage at 24 h post-treatment, 1373 Virus: direct infection of macrophage with 1373 strain (not supernatant), Non-infected: supernatant from macrophage that was not infected, *: *p* < 0.05 (>95% confidence), **: *p* < 0.01 (>99% confidence).

**Figure 9 viruses-12-00701-f009:**
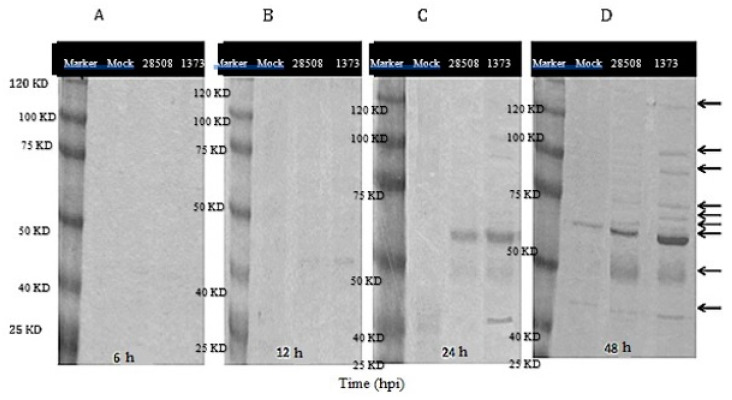
Protein electrophoresis of supernatants from BVDV infected and mock-infected MDM at (**A**) 6 hpi, (**B**) 12 hpi, (**C**) 24 hpi and (**D**) 48 hpi. Marker: protein marker that range from 25 to 120 KD, Mock: non-infected macrophage control, 28508: supernatant from 28508-5-infected macrophage at different timepoints, 1373: supernatant from 1373-infected macrophage at different timepoints, 6, 12, 24, 48 hpi: the corresponding timepoints post treatment of macrophage with supernatant, arrows refer to the different band intensity between strains.

**Table 1 viruses-12-00701-t001:** The set of primers used for quantifications of apoptosis-related cytokine mRNA.

Cytokine	Forward Primer	Reverse Primer	Annealing Temp °C
TNF-α	AGACCCCAGCACCCAGGACTCG	GGAGATGCCATCTGTGTGAGTG	55
IL-1α	GATGCCTGAGACACCCAA	GAAAGTCAGTGATCGAGGG	53
IL-1β	CAAGGAGAGGAAAGAGACA	TGAGAAGTGCTGATGTACCA	53
IL-6	TCCAGAACGAGTATGAGG	CATCCGAATAGCTCTCAG	52
β-actin	CGCACCACTGGCATTGTC	TCCAAGGCGACGTAGCAG	55
IFN-α	ACACACACCTGGT	GATGACAGCAGAAATGA	52
IFN-β	RTCTGSAGCCAAT	CAGGCACACCTGT	52
IFN-γ	ATAACCAGGTCATTCAAAGG	ATTCTGACTTCTCTTCCGCT	55
IL-8	TGGGCCACACTGTGAAAAT	TCATGGATCTTGCTTCTCAGC	53
IL-10	TGCTGGATGACTTTAAGGG	AGGGCAGAAAGCGATGACA	53
IL-12	GAGGCCTGTTTACCACTGGA	CTCATAGATACTTCTAAGGCACAG	58
IL-4	TGCATTGTTAGCGTCTCCTG	AGGTCTTTCAGCGTACTTGT	56

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
