# Peer review of "The Effect of Bovine Viral Diarrhea Virus (BVDV) Strains and the Corresponding Infected-Macrophages’ Supernatant on Macrophage Inflammatory Function and Lymphocyte Apoptosis"

_viruses, 2020, doi:10.3390/v12070701_

Round 1
Reviewer 1 Report
Most of the recommended corrections have been retained and manuscript improved. Taking into account that the results are limited to a very narrow number of strains for a serious comparative study, the current presentation should be modified as short communication. Most of the sections of material and methods and results can be summarized or simply mentioned in the text and presented as supplementary material as figure 7.
Reviewer 2 Report
The manuscript presents the data on BVDV influence on macrophage inflammatory activity and lymphocyte apoptosis. Two ncp strains of BVDV-2a subtype with high and low virulence characteristics were used. The work is presented clearly and all the results are supported by 9 figures accompanying the text. Several minor errors and suggestion for slight modifications are provided below. Brackets in line 86 should be removed or at least the word "ATCC" should be in the regular text. Shortcut "IP" in line 257 should be given in full name although it is given in the caption to figure 1. Caption for figure 5 needs correction since there are no parts A. and B. given in the manuscript. It is suggested to change the words in sentence in lines 341-343 from: ..."had no apoptotic changes" to: ..."did not induce apoptotic changes"... since one can think that supernatant had apoptotic changes. Part C in figure 7 is missing, although the same title is provided for figure 8A. It is suggested to remove word "that" in line 354. Presentation and description of figure 9 is not clear. It is sugested to provide molecular masses for all fragments of marker used in all figures (A to D) since slight shift is visible among those pictures and it is hard to identify the sizes of identified bands. The band described for 1373 strain in fig. B seems to migrate lower at 30kDa position. Therefore addition of marker masses would simlify the interpretation. Shortcut PMNC in lines 436 and 457 should be changed to PBMC or PBMCs. Word "strainis" in line 469 should be changed to "strain is". Word "persistence" in line 471 should be changed to "persistent". Reviewer suggests also to change the word "causes" in line 502 to word "leads" or "prompts".
Author Response
Please see attached file.

This manuscript is a resubmission of an earlier submission. The following is a list of the peer review reports and author responses from that submission.
Round 1
Reviewer 1 Report
Karim Abdelsalam et al report on the effect of BVDV infection on the functional capacity of bovine macrophages. Although I was initially very enthusiastic after reading title and abstract of this manuscript I soon became disappointed. While reading the manuscript I made the following major remarks:
- Introduction: has a literature gap of 4 years. The authors should review the literature from 2016 to 2020 in order to give a complete overview of the state of the art.
Author’s response: Thank you for pointing out this deficiency. We have provided additional references in the background section. - Research question: it is not clear from the state of the art what the researcher’s hypothesis and how their study aims to prove or disprove the hypothesis set forward.
Author’s response: The null hypothesis is that the supernatant from macrophages infected with virulent Ncp BVDV II strain has no effect on non-infected macrophage function and lymphocyte apoptosis. Our research disproves the null hypothesis, that infected macrophage with virulent strain of BVDV lead to lymphocyte apoptosis and macrophage immune dysfunction. This will help to understand the pathogenesis behind the severe lymphoid depletion and general immune dysfunction associated with virulent BVDV infection. - Bifurcation of the content: I have the impression that the effect of BVDV on the functional capacity of macrophages is intertwined with the search for the mode of action of lymphocytopathogenic biotype in pathogenesis of BVDV2. Links also to point 2.
Author’s response: The 1373 strain of BVDV that we used for this proof of concept study was chosen based on the virulence, not the biotype. Our research question was why virulent BVDV infection leads to severe lymphoid depletion and general immune dysfunction in the infected animals. We found that the supernatant of 1373-infected macrophage contains factors that lead to lymphocyte apoptosis and macrophage immune dysfunction. Our results were consistent with the previous literature that indicates that strain 1373 is lymphocytopathogenic. This may explain the apoptotic effect of this strain on lymphocytes and support its effect on macrophage dysfunction. The 1373 strain was chosen because of its virulence not because of its biotype. - Experimental design: as a control a cytopathic strain should have been included cfr. Ridpath Virus Research 118 (2006) 62–69
Author’s response: We used cytopathic 296 C strain of BVDV as a positive control for lymphocyte apoptosis. Since we focus on the supernatant (indirect) than the direct effect, I just did not want to distract or confuse the reader with cp strain. However, I have already added this data to the manuscript (Figure 8A). - Names of strains are not identical to the same strain names used in papers cited.
Author’s response: This has been corrected throughout the manuscript. We used two strains: 1373 and 28508-5. - Results: Although the researcher present a lot of results, many are negative. Which of course is also a results but does raise the question why did you measure this parameter in the first place and why other parameters where not included? This also relates to remark 1. From the introduction it is not clear why authors measured certain parameters and not others. For example they state that the cytokines measured known to induce apoptosis are not produced by macrophages. Although interferon gamma is commonly produced by macrophages but is not measured.
Author’s response: We examined the expression IFN-alpha, beta and gamma but there was no significant difference. This was included in the methods and result sections. Unlike the 2017 Iranian paper that found significant result for IFN-gamma , their research was focused on monocytes not monocyte derived macrophage (MDM). Also, they used different strains than we used and all of this could attribute to the difference in our findings. The reason why I went for cytokines like TNF-alpha, IL-1 beta as they were tested in previous literature of Classical Swine Fever Virus (CSFV) what is the reference here. All other cytokines I used was added to the manuscript (Table 1) - Results: By protein electrophoresis, several protein bands were unique for high virulence 1373-infected MDM supernatant. Why did the authors not identify these bands; could have given novel insight in the pathogenesis.
Author’s response: The reviewer is correct and was also an obvious area of investigation. Our long time proteomics collaborator left the University. We continue to look for a collaborator who can help us with the Mass spectrophotometry and this is on the top of our list to complete. - A critical reflection: Ridpath et al showed in Virus Research 118 (2006) 62–69 that BVDV2-1373 only induced a delayed cell death in BL-3 cells infected with this virus and that the mechanism of cell death was different from cell death induced by a cytopathic biotype. Why did the authors go through all the trouble of isolating bovine macrophages if they could have used BL-3 cell conditioned medium? Did the authors observe the same kinetics for cell death in BL-3 as Ridpath observed in Virus Research 118 (2006)? Was this also the case for MDBK cells? Did the authors fractionate the supernatant of used size exclusion to look into which of the proteins of the conditioned medium on gel induced apoptosis?
Author’s response: Here, and according to the main hypothesis, we want to examine the effect of the infected macrophage on the activity of the neighboring immune cell. According to Pedrera et al., 2009 and Falkenberg et al., 2014 and Liebler-Tenorio et al., 2002, the infected animals with virulent strain of BVDV showed severe lymphoid depletion. This lymphoid depletion was associated with increased number of macrophages 3 days post infection in lymphoid tissues. That’s why we focused on macrophage rather than lymphocytes. Macrophages don’t undergo apoptosis, unlike lymphocytes, upon exposure to virulent BVDV. We did observe the cell death kinetics for cytopathic 296c, and non-cytopathic 1373 and 28508-5 and I have recently added this part to the manuscript (Figure 8A) The pattern of cell death in MDBK and BL-3 was close but not identical as MDBK cells showed more apoptosis % at 48 hours post exposure to 1373-macrophage’s supernatant (Figure 6 and 7). Next step in our research is to perform size exchange chromatography followed by 2D electrophoresis and mass spectrophotometry. We are still seeking a reliable place to perform these 2 tests as we don’t have them at SDSU and hopefully will be done after passing this COVID19 situation.

Reviewer 2 Report
The manuscript is approaching a well known topic. The results are limited to a very narrow number of strains for a serious comparative study. In the discussion, this aspect even if clear should be stated indicating opportunity of further studies to confirm obtained data. However, results, obvious and expected, are based on a study well articulated and merit publication. However, the presentation should be reformulated in a form of short communication. Most of methods section details can be supplementary material, as well as eventually some figures.
Response: Author’s response: Since 1373 is a virulent BVDV II, we choose 28508-5, a mild strain of BVDV II as a comparison. As we are all aware Ncp strains are the primary biotype responsible for all bovine infections . I have included the future studies part in the discussion that point to using other strains in the future work. I appreciate the reviewer’s suggestion for the manuscript to be reformulated as a short communication. Most of the previous literature discussed the lymphoid depletion caused by the infection with virulent Ncp BVDV strains that is caused by lymphoid apoptosis. Since only Cp strains that can induce direct lymphocyte apoptosis, we built our hypothesis that is centered around the Ncp virulent BVDV and how it may induce apoptosis indirectly. On that issue, previous literature has pointed out virus-macrophage interaction but none of them relate it to the lymphocyte apoptosis. Here, our hypothesis suggests a novel way of the induction of lymphocyte apoptosis, upon exposure to virulent Ncp BVDV, through infected macrophage supernatant. For this reason, we are looking for the manuscript to be published as a research article. For the supplementary material point, the reviewer is correct, it is more detailed and took big space of the manuscript. In this situation, I will need guidelines on putting this detailed part in a separate supplementary file without affecting the method section in the manuscript body.
A number of corrections are necessary
Key words in alphabetic order
Author's response: The key words were reorganized alphabetically.
Page 1 line 34 : delete dot after “pathogenesis”
Author's response: The dot was removed after pathogenesis.
Page 2 line 65 : add reference Mangan et al., 1992 in the reference list, renumber accordingly and insert related number in the text.
Author's response: The reference was added to the reference section and the corresponding reference number was added appropriately.
Do the same for references “Blond et al., 2000” (Page 4 line 151), “CELL Quest software” (Page 4 line 174), “Mi Hyeon et al., 2002” (Page 4 line 187) and “Schagger and von Jagow, 1987” (Page 5 line 207)
Author's response: The 3 references were added with corresponding new numbered citations corrected throughout the manuscript. For the software “Cell Quest”, I put the source in the main text as previously cited in other papers. An example of this citation is demonstrated in this paper
in the section labeled “Phagocytic Assays”: https://rupress.org/jcb/article/164/2/185/33528/The-uniformity-of-phagosome-maturation-in?fbclid=IwAR02lUyKwci5WSRfzBCaJbT9cQpepVJxEvuqyXnZySDtgHi0P-72VmpOoNk.
Include always full manufacturer details, often USA is omitted (for example: Page 2 lines 77 and 78) see various others.
Author's response: Thank you. The method section has been extensively edited and the manufacturer details was added throughout the section.
Page 2 line 85: “monocyte derived macrophage (MDM)” instead of “MDM (monocyte derived macrophage)”
Author's response: This has been changed as suggested..
Page 2 line 89: add full word followed by acronym “room temperature (RT)” instead of “RT” and delete “room temperature” at Page 5 line 213
Author's response: This has been changed throughtout the entire manuscript, to make sure that room temperature (RT) was only at page 2 line 89 when first introduced then just putting RT in the following positions throughout the article.
If needed, include the name of author followed by reference number; for example: Page 2 line 91: “according to Strober [47].” instead of “according to [47].”
Author's response: This has been changed successfully as recommended for the above example as well as page 4 line 182 and Page 5 line 227.
Add reference number [12] and delete year for citation of Elmowalid Page 3 lines 97 and 104: “Elmowalid [12]” instead of “Elmowalid, 2003” see various others as Page 4 line 181 or Page 5 line 226
Author's response: This has been changed. The other 2 occurrences and others have been edited already in the previous comment.
Page 3 line 140 : add manufacturer specifications for “ELISA reader”
Author's response: This was added.
Page 4 line 153 and 156: complete manufacturer specifications for “Sigma”
Author's response: This was added.
Page 4 line 162: add manufacturer specifications for “AccutaseTM”
Author's response: This was successfully added.
Page 4 line 173: “mean” instead of “Mean” not capital
Author's response: This has been corrected.
Page 5 line 203: “AccuriTM” instead of “AccuriTM”
Author's response: This has been corrected.
Page 5 line 210: “100oC” instead of “100oC”
Author's response: This has been corrected.
Page 5 line 217: add NA after “nucleic acid”
Author's response: This has been corrected throughout the article.
Page 5 line 225: delete double comma after “Scientific”
Author's response: This has been corrected.
Page 6 line 235: “kindly” instead of “Kindly” not capital
Author's response: This has been corrected.
Pages 6-7 lines 243-245: delete “and was indicated on the graphs by one asterisk” and “that was represented by 2 asterisks”
Author's response: This has been deleted.
Page 7 line 258 : “strain (Figure 1C) compared to the non-irradiated infection control (Figure 1D).” instead of “strain compared to the non-irradiated infection control (Figure 1C,1D).”
Author's response: This sentence has been edited as recommended.
Page 8 Figure 2 legend : Add meaning for “V”; Supernatant and Macrophage not in capital; “ *: p<0.05 (>95% confidence).” instead of “* indicates significant difference with more than 95% confidence (p<0.05).”
Author's response: The entire legend was edited based on your recommendations.
Page 8 line 289 : dot after “(Figure 3)”
Author's response: This has been corrected
Page 11 Figure 6: part B details not clear and missing ** statistic symbols; legend: see note for figure 2.
Author's response: Part B details were successfully added. Part B is the explanation of A but in %. The legend details has been corrected.
Page 12 Figure 7: part B insufficient resolution quality and details not clear; legend: see note for figure 2.
Author's response: Unfortunately, this is the highest resolution available with this image. We can try enhancing the image using an image editor to enhance the quality, but not to modify anything. Otherwise the Figure could be removed from the manuscript and just referred to as “data not shown” in the results and discussion sections.
Page 13 Figure 8: add colors to line names; missing (p<0.01) ** statistic symbol; legend: see note for figure 2.
Author's response: The color has been added correspondingly. There was no ** significance so it was initially added mistakenly and I have deleted from the legend.
In result section delete redundant description of methods section or complete methods section:
Author's response: This has been done as you recommended.
Page 7 lines 251-255: delete “The infected MDM supernatant was UV-irradiated to inactivate infectious BVDV particles to exclude the direct effect of the virus. To determine that the BVDV virus was inactivated by irradiation, BVDV-susceptible MDBK cells were treated with irradiated supernatant for an hour and the infected cells were stained for BVDV antigen expression in an immunoperoxidase (IP) assay in addition to direct fluorescent assay (DFA) to detect virus replication.”
Author's response: This has been deleted.
Page 7 lines 268-270: delete “Irradiated supernatant from BVDV-infected MDM was used to study the indirect effect of BVDV on the phagocytic activity of the non-infected macrophages at different time points post treatment.”
Author's response: This has been deleted.
Page 8 lines 286-287: delete “Virus-free supernatant from BVDV-infected macrophage was used to study the bactericidal activity of macrophage at different time points post treatment.”
Author's response: This has been deleted.
Page 9 lines 302-303: delete “We investigated the effect of BVDV-infected MDM supernatants on nitric oxide production of uninfected MDM in comparison with the BVDV-infected MDM.”
Author's response: This has been deleted.
Page 10 lines 314-315: delete “MHC II expression was also investigated after treatment of uninfected cells with supernatant from Ncp 1373 or 28508 infected MDM.”
Author's response: This has been deleted.
Page 10 lines 321-322: delete “The effect of infected MDM supernatant as well as direct BVDV infection on macrophage surface marker expression was measured at different time points.”
Author's response: This has been deleted.
Page 11 lines 335-337: delete “To study the apoptotic effect of BVDV-infected MDM supernatant, we treated MDBK cells with 1373 or 28508 virus free, 24h-infected MDM supernatant for up to 48 h. The percentage of apoptotic cells was measured using chromatin condensation and Annexin V staining.”
Author's response: This has been deleted.
Page 12 lines 352-353: delete “BVDV-infected macrophages with either 1373 or 28508 strains were used to treat fresh peripheral blood lymphocytes or BL-3 cell line.”
Author's response: This has been deleted.
Page 13 lines 377-379: delete “To investigate the difference between BVDV strain supernatants, the supernatants were analyzed using gel electrophoresis. The MDM cells were infected with BVDV in serum free media for 6, 12, 24 or 48 h.”
Author's response: This has been deleted.
Page 14 lines 392-393: delete “Bovine cytokine mRNA analysis was performed by quantitative RT-PCR on the infected MDM lysate at 3 different time points; 12, 24 and 48 h post infection for” and “. These”
Author's response: This has been deleted and the cytokines have been included in the text.
Page 14 lines 398-403: delete “To investigate if BVDV viral factors are responsible for the indirect lymphocyte apoptosis, either 15c5 Erns specific monoclonal antibody (mAb) (IDEXX Laboratories, Westbrook, ME, USA) or a BVDV-specific polyclonal antibody, kindly provided by Dr. Robert Fulton, were used to neutralize the supernatant of infected MDM of the Ncp 1373 strain of BVDV. Following neutralization, the BL-3 cells were treated with the neutralized infected supernatant 1373 strains and the percentage of lymphocyte apoptosis was calculated by flow cytometry.”
Author's response: This has been deleted.